# Two-dimensional ferroelasticity in van der Waals $\beta'$-In$_2$Se$_3$

Chao Xu[1], Jianfeng Mao[1], Xuyun Guo [1], Shanru Yan[1], Yancong Chen[2], Tsz Wing Lo[1], Changsheng Chen[1], Dangyuan Lei [3], Xin Luo[2], Jianhua Hao [1], Changxi Zheng[4,5] & Ye Zhu [1✉]

Two-dimensional (2D) materials exhibit remarkable mechanical properties, enabling their applications as flexible and stretchable ultrathin devices. As the origin of several extraordinary mechanical behaviors, ferroelasticity has also been predicted theoretically in 2D materials, but so far lacks experimental validation and investigation. Here, we present the experimental demonstration of 2D ferroelasticity in both exfoliated and chemical-vapor-deposited $\beta'$-In$_2$Se$_3$ down to few-layer thickness. We identify quantitatively 2D spontaneous strain originating from in-plane antiferroelectric distortion, using both atomic-resolution electron microscopy and in situ X-ray diffraction. The symmetry-equivalent strain orientations give rise to three domain variants separated by 60° and 120° domain walls (DWs). Mechanical switching between these ferroelastic domains is achieved under ≤0.5% external strain, demonstrating the feasibility to tailor the antiferroelectric polar structure as well as DW patterns through mechanical stimuli. The detailed domain switching mechanism through both DW propagation and domain nucleation is unraveled, and the effects of 3D stacking on such 2D ferroelasticity are also discussed. The observed 2D ferroelasticity here should be widely available in 2D materials with anisotropic lattice distortion, including the 1T' transition metal dichalcogenides with Peierls distortion and 2D ferroelectrics such as the SnTe family, rendering tantalizing potential to tune 2D functionalities through strain or DW engineering.

[1] Department of Applied Physics, Research Institute for Smart Energy, The Hong Kong Polytechnic University, Hung Hom, Kowloon, Hong Kong, China. [2] State Key Laboratory of Optoelectronic Materials and Technologies, Centre for Physical Mechanics and Biophysics, School of Physics, Sun Yat-sen University, Guangzhou, China. [3] Department of Materials Science and Engineering, City University of Hong Kong, 83 Tat Chee Avenue, Hong Kong, China. [4] School of Science, Westlake University, Hangzhou, China. [5] Institute of Natural Sciences, Westlake Institute for Advanced Study, Hangzhou, China. ✉email: yezhu@polyu.edu.hk

Van der Waals (vdW) layered materials have attracted tremendous research efforts to explore emerging physics at reduced dimensionality[1–4]. The intrinsically weak interlayer interaction allows each individual layer to behave independently, granting the unique opportunity to preserve bulk properties down to single-layer thickness[5–7]. Various two-dimensional (2D) functionalities have been discovered in vdW materials[7], including superconductivity[8–10], ferromagnetism[11,12], ferroelectricity[13–17] and antiferroelectricity[18,19], offering a wealth of choices for making ultrathin flexible 2D heterostructure devices[7,20]. Among all the ferroic properties, ferroelasticity represents the mechanical equivalent of ferromagnetism and ferroelectricity, with multiple orientation states of spontaneous lattice strain that are switchable under mechanical stimuli[4,21,22]. It is the origin of the shape-memory effect and superelasticity[23,24], relevant to applications including piezoelectric sensors, mechanical switches, and actuators[4,25–27]. 2D ferroelasticity has also been predicted in monolayer 1T' transition metal dichalcogenides (TMDs) based on first-principles calculations[28,29], which may couple with other extraordinary properties to realize strain-tunable functionalities and 2D multiferroics[4]. However, despite several proposed 2D ferroelastic behavior[28–31] and one very recent demonstration on micron-thick layered perovskites[32], 2D ferroelasticity evidenced by the mechanical switch of spontaneous lattice strain in ultrathin vdW materials has not yet been validated experimentally[4].

Here we report the experimental demonstration of 2D ferroelasticity in vdW $\beta$'-In$_2$Se$_3$ down to few-layer thickness. As a III$_2$-VI$_3$ compound semiconductor, In$_2$Se$_3$ is polymorphic with many phases reported ($\alpha$', $\alpha$, $\beta$', $\beta$, $\gamma$, $\delta$, $\kappa$)[33–38]. Among them, the two room-temperature phases, $\alpha$- and $\beta$'-In$_2$Se$_3$, and the high-temperature $\beta$ phase are vdW materials with 2D [Se-In-Se-In-Se] quintuple layers stacked in several possible ways (2H, 3R, 1T)[35,39–41]. At room temperature, $\alpha$-In$_2$Se$_3$ exhibits 2D ferroelectricity that is currently under intensive investigation[15,38,42–46]. $\beta$'-In$_2$Se$_3$, on the other hand, possesses the characteristic superstructure consisting of periodic nanostripes[35,39], the nature of which was only clarified recently as 2D antiferroelectricity competing with the ferroelectric ordering[18]. It makes In$_2$Se$_3$ a fascinating system for exploring both the fundamental ferroelectric physics at the 2D limit and ultrathin phase-switching device applications[42,43,45–49]. In this work, we identify quantitatively the 2D spontaneous strain originating from in-plane antiferroelectric distortion in $\beta$'-In$_2$Se$_3$, using a combination of atomic-resolution scanning transmission electron microscopy (STEM) and in situ X-ray and electron diffraction. Polarized-light microscopy further unveils multi-domain structure in $\beta$'-In$_2$Se$_3$ with three strain-orientation variants that are energy degenerate. Moreover, the switch between the domain variants is demonstrated unequivocally by applying moderate external strain. The observed ferroelasticity can be preserved down to few-layer thickness, which proves unambiguously its 2D nature.

## Results

### Thermoelastic transformation and 2D spontaneous strain in $\beta$'-In$_2$Se$_3$.
$\beta$'-In$_2$Se$_3$ is distinguished from the parent high-temperature $\beta$ phase by its characteristic nanostriped superstructure (Fig. 1a), whose antiferroelectric nature is reflected by the antiparallel Se displacement between the neighboring nanostripes (Fig. 1b and also ref. [18]). Such nanostriped superstructure can also be detected by the satellite diffraction in electron diffraction patterns (Fig. 1c). Other than that, the main diffraction from $\beta$- and $\beta$'-In$_2$Se$_3$ is largely the same, indicating their nearly identical basic structure. The structure of $\beta$'-In$_2$Se$_3$ can thus be understood as the parent $\beta$-In$_2$Se$_3$ structure modified by the nanostriped superstructure with antiferroelectric

ordering[18]. In this manuscript, crystallographic indexing refers to the parent hexagonal lattice rather than the superstructure.

Using in situ heating/cooling X-ray diffraction (XRD), we further detect subtle lattice distortion caused by antiferroelectricity in $\beta$'-In$_2$Se$_3$. Compared to the high-temperature $\beta$ phase with higher symmetry[18,41], the room-temperature $\beta$' phase exhibits numerous peak splitting as shown in Supplementary Fig. 1, indicating anisotropic lattice distortion and the associated symmetry breaking. The structure refinement indeed shows spontaneous lattice dilation along the nanostripes and compression perpendicular to the nanostripes in $\beta$'-In$_2$Se$_3$ associated with the nonpolar-to-polar structure transition, as illustrated quantitatively by the temperature dependence of lattice spacing plotted in Fig. 1d. Using $\sqrt{3}d^{\parallel}_{11\bar{2}0}$ and $d^{\perp}_{1\bar{1}00}$ to represent the lattice spacing parallel and perpendicular to the nanostripes respectively (also indicated in Fig. 1b), the hexagonal lattice of the $\beta$ phase requires $d^{\perp}_{1\bar{1}00} = \sqrt{3}d^{\parallel}_{11\bar{2}0}$ as observed above 250 °C. By lowering temperature to initiate $\beta$-to-$\beta$' transition, both an increase of $\sqrt{3}d^{\parallel}_{11\bar{2}0}$ and a decrease of $d^{\perp}_{1\bar{1}00}$ are detected. This transition is fully reversible as evidenced by the in situ heating curve also plotted in Fig. 1d. Adopting Aizu's definition of strain[50,51] and assigning [$1\bar{1}00$] and [$11\bar{2}0$] as $x$ and $y$ axes respectively, as shown in Fig. 1b, the 2D spontaneous strain tensor can be derived as

$$\boldsymbol{\varepsilon}(A) = \begin{pmatrix} \varepsilon_{xx} & \varepsilon_{xy} \\ \varepsilon_{xy} & \varepsilon_{yy} \end{pmatrix} = \begin{pmatrix} -0.0049 & 0 \\ 0 & 0.0049 \end{pmatrix}, \qquad (1)$$

where $\varepsilon_{xx}$ and $\varepsilon_{yy}$ are the tensile or compressive strain along $x$ and $y$ directions, both about 0.0049 for room-temperature $\beta$'-In$_2$Se$_3$, and $\varepsilon_{xy}$ is the shear strain component. It is noted that ~0.49% spontaneous strain is much smaller than the strain values reported in 1T' TMDs[28,52], which are typically a few percent.

**Multidomain variants and domain walls in $\beta$'-In$_2$Se$_3$.** Owing to the hexagonal lattice of the parent structure, the nanostripes and the associated lattice strain in $\beta$'-In$_2$Se$_3$ can orient along one of the three symmetry-equivalent <$11\bar{2}0$> directions ([$11\bar{2}0$], [$1\bar{2}10$] and [$\bar{2}110$]) and thus give rise to three distinct domain variants. The change of nanostripe orientation between the neighboring domains is clearly revealed by both atomic-resolution STEM images (Fig. 2a, b) and electron diffraction (Fig. 2c, d). Besides showing the two directions of nanostripe ordering from the satellite diffraction, electron diffraction in Fig. 2c, d also displays 'split spots' especially for high-order diffraction farther away from the twin axis. This is because the diffraction spots from different domains are slightly separated, due to their distinct spontaneous strain orientation. As illustrated by the schematic model in Fig. 2f, with the nanostripes changing their orientation across a domain wall (DW), the spontaneous strain changes direction accordingly and causes a small deviation angle between the lattices of the two domains. With the tensile and compressive strain both ~0.49% measured from XRD, a deviation angle of ~0.97° can be derived which matches the separation angle between the 'split spots' in diffraction patterns. The strain-orientation variants can also be revealed directly in real space, through lattice spacing mapping on atomic-resolution images such as Fig. 2b. As shown in Fig. 2e, the upper domain shows clearly the larger horizontal lattice spacing of $d_{1\bar{1}00}$ compared to the lower domain, owing to the change of strain orientation as illustrated in Fig. 2f.

All three domain variants in $\beta$'-In$_2$Se$_3$ can be visualized at a larger scale by polarized-light imaging shown in Fig. 3a. The domain contrast arises from the linear-dichroism behavior as depicted by angular-resolved polarized-light imaging in Fig. 3b. As the domain intensity is modulated by the orientation of nanostripes

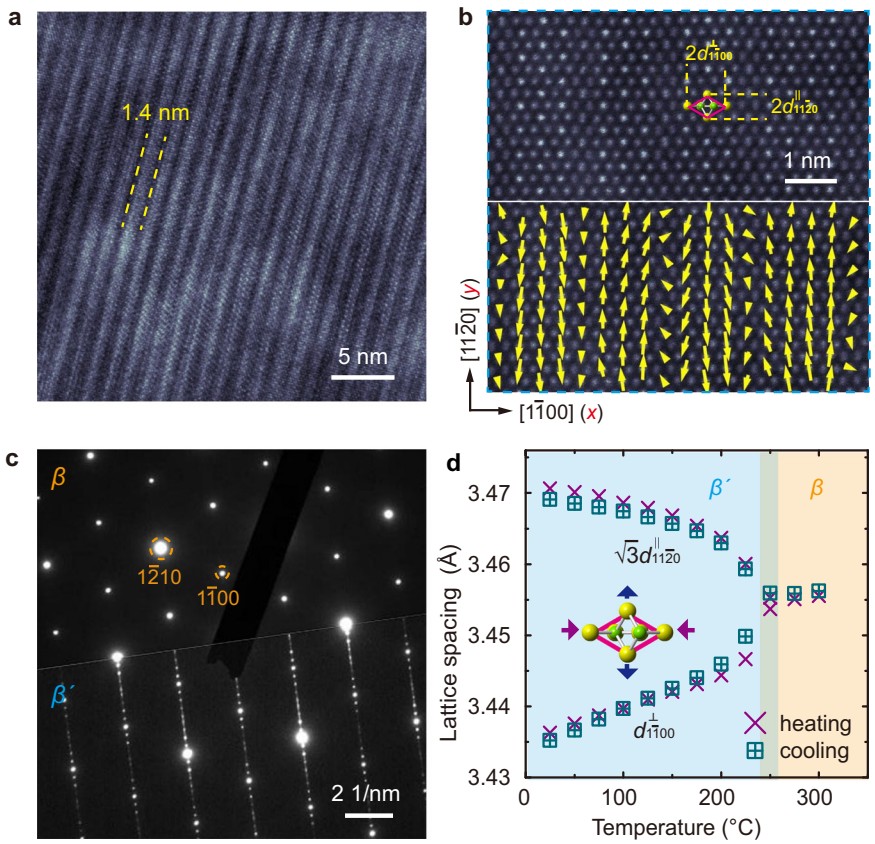

**Fig. 1 Antiferroelectricity-coupled spontaneous strain in $\beta'$-In$_2$Se$_3$. a** TEM image showing the characteristic nanostriped superstructure in 2H $\beta'$-In$_2$Se$_3$. **b** Atomic-resolution annular dark-field (ADF) STEM image of the nanostripes and the atomic displacement map showing antiferroelectric ordering. **c** Selected-area electron diffraction (SAED) patterns of the parent $\beta$ (top) and the room-temperature $\beta'$ (bottom) phases. The nanostriped superstructure in $\beta'$-In$_2$Se$_3$ gives rise to satellite diffraction at $n/8$ $1\bar{1}00$ corresponding to the characteristic nanostripe width of $4d_{1\bar{1}00}$ (~1.4 nm). **d** In-plane lattice spacing as a function of temperature extracted from in situ XRD for both heating and cooling processes. The observed increase of $\sqrt{3}d^{\parallel}_{11\bar{2}0}$ and decrease of $d^{\perp}_{1\bar{1}00}$ during $\beta$-to-$\beta'$ transition reflect the anisotropic spontaneous strain with lattice dilation (compression) along (perpendicular to) the $\beta'$-In$_2$Se$_3$ nanostripes.

relative to the light polarization direction, the strain orientation in each domain variant can then be easily determined. Figure 3b evinces clearly the angular difference of 60°/120° between different domain variants, consistent with the domain structure shown in Fig. 2. Denoting the three domain variants as A, B and C, whose tensile strain (or nanostripe) directions are indicated in Fig. 3a, the spontaneous strain tensors $\boldsymbol{\varepsilon}(B)$ and $\boldsymbol{\varepsilon}(C)$ can then be derived by transforming $\boldsymbol{\varepsilon}(A)$ using the rotation matrix

$$\mathbf{J}_{\pm} = \begin{pmatrix} \cos(\pm 120°) & -\sin(\pm 120°) \\ \sin(\pm 120°) & \cos(\pm 120°) \end{pmatrix}, \qquad (2)$$

with

$$\boldsymbol{\varepsilon}(B) = \mathbf{J}_{+}\boldsymbol{\varepsilon}(A)\mathbf{J}_{+}^{-1} = \begin{pmatrix} 0.0025 & 0.0042 \\ 0.0042 & -0.0025 \end{pmatrix}, \qquad (3)$$

$$\boldsymbol{\varepsilon}(C) = \mathbf{J}_{-}\boldsymbol{\varepsilon}(A)\mathbf{J}_{-}^{-1} = \begin{pmatrix} 0.0025 & -0.0042 \\ -0.0042 & -0.0025 \end{pmatrix}. \qquad (4)$$

$\boldsymbol{\varepsilon}(A)$, $\boldsymbol{\varepsilon}(B)$ and $\boldsymbol{\varepsilon}(C)$ form a complete set of spontaneous strain tensors that are consistent with Aizu's 3D tensors for $6\,mm-mm2$ or $\bar{3}m-2/m$ ferroelastic-transition species[21].

The distinct domain variants in $\beta'$-In$_2$Se$_3$ are separated by long straight DWs as shown in Fig. 3a. Both 60° and 120° DWs can form between each pair of domain variants (A-B, B-C, A-C), as indicated in the bottom inset of Fig. 3a and also imaged at atomic scale in Fig. 2a, b for the A-B pair. Despite the orientation change of the nanostripes in Fig. 2a, b, the basic lattice of $\beta'$-In$_2$Se$_3$

remains coherent across both DWs. As required by the mechanical compatibility of coherent DWs[51,53], 60° DWs in $\beta'$-In$_2$Se$_3$ must align along $\{1\bar{1}00\}$ planes, while 120° DWs must be parallel to $\{11\bar{2}0\}$ planes (see Fig. 3a and Supplementary Note 1), which explains their long straight shape and the orthogonal grid pattern observed in Fig. 3a. As $\{11\bar{2}0\}$ planes are mirror planes of the parent structure while $\{1\bar{1}00\}$ planes are not, this classifies 120° DWs as $W_f$ walls whose orientations are fixed by the lattice symmetry, and 60° DWs as $S$ walls whose orientations depend on the spontaneous strain[51].

**Ferroelastic domain switching by external strain.** To further validate ferroelasticity, we demonstrate explicitly domain switching by applying uniaxial tensile strain to the multidomain $\beta'$-In$_2$Se$_3$ flakes. As shown in Fig. 4a, the pristine flake shows two stable domain variants A and B whose spontaneous lattice-dilation directions are indicated by double-headed arrows. By gluing it onto a flexible substrate and bending it upward, we then stretch the flake with external tensile strain whose value can be quantitatively measured (see Methods)[54]. Under such external vertical tension, the darker A domains with the vertical lattice dilation become energetically more favorable and thus grow at the expense of the B domains. This is mostly accomplished through the progressive propagation of existing DWs such as those indicated by the dashed lines in Fig. 4a, b. The nucleation of A domains inside B is also observed in the circled regions in Fig. 4b. Further increasing tensile strain eventually leads to the switch of

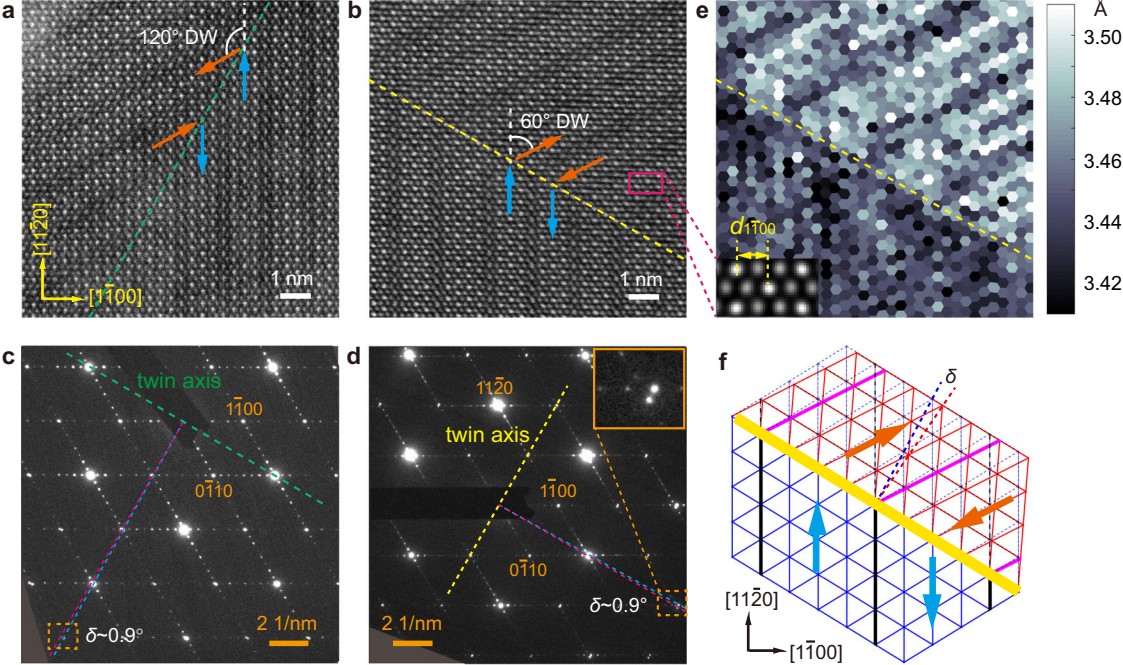

**Fig. 2 Multidomain and DW structure in β'-In₂Se₃. a, b** Atomic-resolution ADF-STEM images of 120° and 60° DWs in 2H β'-In₂Se₃, revealing the unchanged basic structure but the switch of the nanostripe directions across the coherent DWs. **c, d** SAED patterns of the 120° and 60° DWs. Diffraction spots not on the twin axes show 'split spots' as magnified in the top-right inset of **d**. **e** Horizontal lattice spacing (as defined in the bottom-left inset) mapped around the 60° DW shown in **b**, reflecting lattice dilation along the nanostripes. **f** Schematic of the 60° DW structure based on SAED pattern in **d**, which can explain the atomic structure and lattice distortion imaged in **b** and **e**. The original lattice without twinning is drawn in dashed thin blue lines, which reveal a deviation angle δ that matches the separation angle of diffraction spots in **d**. The antiparallel polarizations of neighboring nanostripes are indicated by blue/orange arrows in **a**, **b** and **f**.

nearly all B domains to A (Fig. 4d), representing the lowest-energy configuration of lattice distortion in response to the applied strain. Unloading the external strain leads to a reverse switch of A domains to B. However, even when the applied strain is completely released, the final domain pattern in Fig. 4g does not fully recover to its pristine state: There are more A domains remaining compared to Fig. 4a, especially for those newly formed A domains through nucleation, reflecting the hysteretic nature of ferroelasticity[55].

Using the domain area fraction of A variant as an indicator[55], we can describe quantitatively the ferroelastic switching behavior as plotted in Fig. 4h. The fastest growth of A domains occurs under the intermediate strain (0.5% ≤ ε ≤ 1.1%), suggesting that the applied strain in this range is mostly accommodated by domain switching. Indeed, the B-to-A switching can give rise to ~0.74% elongation along the vertical direction (see Supplementary Note 2). Assuming all the applied strain is accommodated by this domain switching, a linear response is predicted along the blue dotted line in Fig. 4h (Supplementary Note 2), which matches the experimental observation and validates our interpretation. On the other hand, the slower domain switching outside this linear regime may be due to some domain-wall pinning mechanism, which needs further investigation. We have explicitly excluded the potential phase change or flexoferroelectric effect during strain-induced domain switching, using in situ Raman spectroscopy and second harmonic generation (see Supplementary Fig. 3). It is worth noting that the external strain as small as 0.5% can induce domain switching in β'-In₂Se₃, which is considerably smaller than the switching strain (≥1%) predicted in 1T' TMDs by first-principles calculations[28]. The yield strain of In₂Se₃ flakes is determined to be ~5.5% by atomic force microscopy (AFM) nano-indentation (see Supplementary Fig. 4 and Note 3), much larger than the domain-switching strain.

Another example of ferroelastic domain switching involving all three domain variants is presented in Supplementary Fig. 5. The initial state is composed of A-C variants with the presence of both 60° and 120° DWs. With the external tensile strain applied horizontally, A variant with vertical lattice dilation becomes unfavorable, which leads to both the growth of existing C domains by DW propagation and the nucleation of B and C domains inside A (Supplementary Fig. 5b). Increasing tensile strain makes the newly formed domains propagate both along and perpendicular to the DWs, until all the A domains are switched in Supplementary Fig. 5e. Switching between B and C variants is also observed, as indicated in Supplementary Fig. 5c, d.

We note that the ferroelastic domain switching demonstrated here is underpinned by the spontaneous 2D strain associated with the antiferroelectric distortion and nanostriped superstructure formation. First-principles calculation has shown that the antiferroelectric distortion as well as the superstructure can be preserved down to monolayer β'-In₂Se₃[18], which immediately suggests the preservation of ferroelasticity at this 2D limit. To experimentally demonstrate such 2D ferroelasticity, we adopt chemical vapor deposition (CVD) to grow ultrathin β'-In₂Se₃ flakes, which offer larger flat area for imaging multidomains compared to mechanical exfoliation. As seen in Fig. 5a, ferroelastic domains are still visible in a CVD-grown β'-In₂Se₃ flake as thin as ~6.6 nm (7 quintuple layers), measured by AFM shown in Fig. 5b. Similar domain patterns have also been observed in thinner flakes with ~5.7 nm thickness (Supplementary Fig. 6). Electron diffraction in Fig. 5c further shows the presence of nanostriped superstructure in such ultrathin flakes as evidenced by the corresponding satellite diffraction. More excitingly, ferroelastic domain switching similar to Fig. 4 can also be achieved on ultrathin β'-In₂Se₃ flakes, as demonstrated in Fig. 5d, f, which proves unequivocally the 2D nature of the

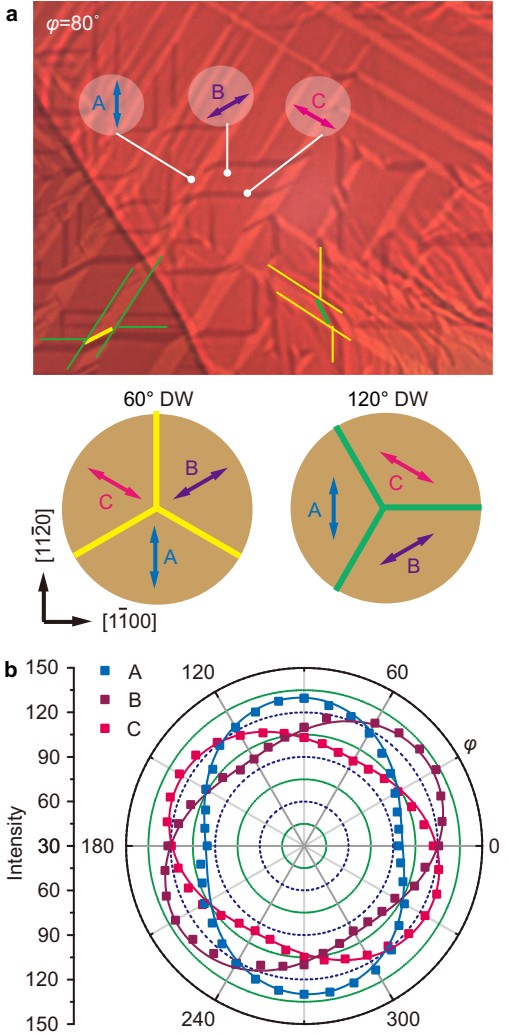

**Fig. 3 Angular-resolved polarized-light imaging of multidomain pattern in $\beta'$-In$_2$Se$_3$. a** Polarized-light image at the polarizer azimuth angle $\varphi = 80°$ presenting the typical multidomain pattern in 2H $\beta'$-In$_2$Se$_3$, with all permissible domain variants and DWs depicted at the bottom. **b** Azimuth-dependent transmission intensity of the three domain variants A, B and C. The colored squares show the measured data that can be well described by cosine functions, indicating linear dichroism. The antiferroelectric polarization (or nanostripe) direction in each domain variant is determined to be along the light polarization direction where the transmission intensity is maximal.

observed ferroelasticity. Even though ferroelasticity has been predicted theoretically in other 2D materials such as monolayer 1T' TMDs[28,29], to our best knowledge, this is the first experimental study that systematically demonstrates 2D ferroelasticity in ultrathin vdW materials.

**Ferroelasticity tailored by stacking ordering in $\beta'$-In$_2$Se$_3$.** The analysis above focuses on 2D ferroelasticity only, which solely depends on the atomic structure within the $\beta'$-In$_2$Se$_3$ quintuple layer. However, when extending to 3D, the 2D ferroelasticity can be further affected by the quintuple-layer stacking ordering, with 2H and 3R $\beta'$-In$_2$Se$_3$ manifesting different symmetry and ferroelastic behavior (Fig. 6a, b). Compared to 2H $\beta'$-In$_2$Se$_3$ with about equal population of 60° and 120° DWs (Fig. 6a), 3R $\beta'$-In$_2$Se$_3$ is dominant by 60° DWs with fewer 120° DWs observed (Fig. 6b and Supplementary Fig. 7). More strikingly, across those

minor 120° DWs in 3R $\beta'$-In$_2$Se$_3$, unusual surface wrinkles are detected by AFM as depicted in Fig. 6b, which are not observed in 2H $\beta'$-In$_2$Se$_3$. This indicates the presence of shear strain with the out-of-plane component in 3R $\beta'$-In$_2$Se$_3$, but not in 2H $\beta'$-In$_2$Se$_3$. Indeed, for 3R $\beta'$-In$_2$Se$_3$ with a monoclinic lattice, its [0001] direction is not perpendicular to the in-plane [1$\bar{1}$00] direction but slightly inclined with an angle of 91.44°, as consistently identified by both XRD structure refinement and cross-sectional STEM imaging (Fig. 6b). It gives rise to the shear strain long [1$\bar{1}$00] within the (0001) plane, *i.e.*, the $\varepsilon_{xz}$ component in the strain tensors in addition to the above in-plane 2D strain. Adding [0001] as $z$ axis to the previous 2D strain tensor, we obtain the 3D tensor for the A variant in 3R $\beta'$-In$_2$Se$_3$ as

$$\boldsymbol{\varepsilon}(\text{A}, 3\text{R}) = \begin{pmatrix} \varepsilon_{xx} & \varepsilon_{xy} & \varepsilon_{xz} \\ \varepsilon_{xy} & \varepsilon_{yy} & \varepsilon_{yz} \\ \varepsilon_{xz} & \varepsilon_{yz} & \varepsilon_{zz} \end{pmatrix} = \begin{pmatrix} -0.0049 & 0 & -0.0126 \\ 0 & 0.0049 & 0 \\ -0.0126 & 0 & 0 \end{pmatrix} \quad (5)$$

with the shear strain $\varepsilon_{xz} = -0.0126$ as derived from the 91.44° inclined angle. The 3D strain tensors for B and C variants can be derived by transforming $\boldsymbol{\varepsilon}(\text{A})$ using the rotation matrix (see Supplementary Note 4) as

$$\boldsymbol{\varepsilon}(\text{B}, 3\text{R}) = \begin{pmatrix} 0.0025 & 0.0042 & 0.0063 \\ 0.0042 & -0.0025 & -0.0109 \\ 0.0063 & -0.0109 & 0 \end{pmatrix}, \quad (6)$$

$$\boldsymbol{\varepsilon}(\text{C}, 3\text{R}) = \begin{pmatrix} 0.0025 & -0.0042 & 0.0063 \\ -0.0042 & -0.0025 & 0.0109 \\ 0.0063 & 0.0109 & 0 \end{pmatrix}. \quad (7)$$

When two variants meet at 120° DWs in 3R $\beta'$-In$_2$Se$_3$, their shear strain components perpendicular to DWs possess the opposite signs, as graphically represented in the inset of Fig. 6d. To maintain a coherent lattice at the DWs, the mosaic tilt of the (0001) planes results and manifests as surface wrinkles (see the middle panel of Fig. 6c). Such a mosaic tilt has been detected explicitly through reciprocal space mapping (RSM) that shows separate out-of-plane diffraction from the distinctly titled (0001) planes in A and B/C domains (Fig. 6d). When two variants meet at 60° DWs, on the other hand, their shear strain components perpendicular to DWs share the same sign, which leads to a flat surface without wrinkles (see Supplementary Fig. 8a and its inset) as confirmed by AFM measurement (Fig. 6b). We emphasize that the surface wrinkles and the underlying shear strain only exist in monoclinic 3R $\beta'$-In$_2$Se$_3$. In contrast, both 60° and 120° DWs in 2H $\beta'$-In$_2$Se$_3$ exhibit flat surfaces as revealed by AFM (Fig. 6a), owing to its orthorhombic lattice resulting from the reversed AB'AB' stacking of the quintuple layers (Fig. 6a). Therefore the spontaneous strain in 2H $\beta'$-In$_2$Se$_3$ is purely 2D without any out-of-plane component (see Supplementary Note 4). All observation of 2D ferroelasticity in previous sections are thus based on 2H $\beta'$-In$_2$Se$_3$ instead of 3R $\beta'$-In$_2$Se$_3$.

Besides the wrinkled surface across 120° DWs, the shear strain in 3R $\beta'$-In$_2$Se$_3$ also leads to inclined 60° DWs as required by the mechanical compatibility[51,53] for coherent DWs (Supplementary Fig. 8 and Supplementary Table 2). Such inclined 60° DWs are indeed observed via both polarized-light microscopy and piezo-response force microscopy (PFM). As shown in Fig. 7b, across a step edge with the height ~270 nm, all the 60° DWs exhibit a horizontal shift of ~480 nm, reflecting their inclined nature. We note that the observed domain contrast in lateral PFM phase image (Fig. 7b) is attributed to the second-order electrostrictive effect, and thus the phase difference between domains is less than 180° (see Supplementary Note 5). Like the surface wrinkles, such

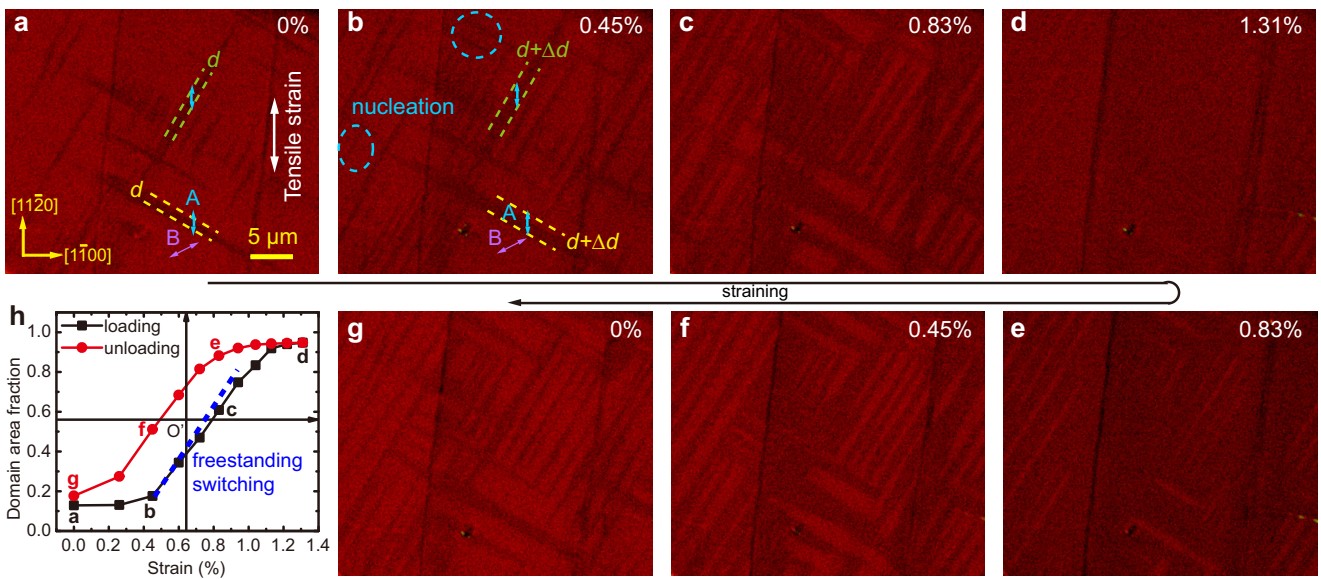

**Fig. 4 Ferroelastic domain switching in an exfoliated $\beta'$-In$_2$Se$_3$ flake. a–g** Polarized-light images showing domain switching under the vertical uniaxial tensile strain. The colored arrows indicate the nanostripe or antiferroelectric polarization directions. **h** Domain area fraction of the A variant as a function of applied uniaxial tensile strain. The linear response under the intermediate strain (0.5% $\leq \varepsilon \leq$ 1.1%) can be well described by the blue line with a slope of 1.35 as derived from a freestanding switching model presented in Supplementary Note 2.

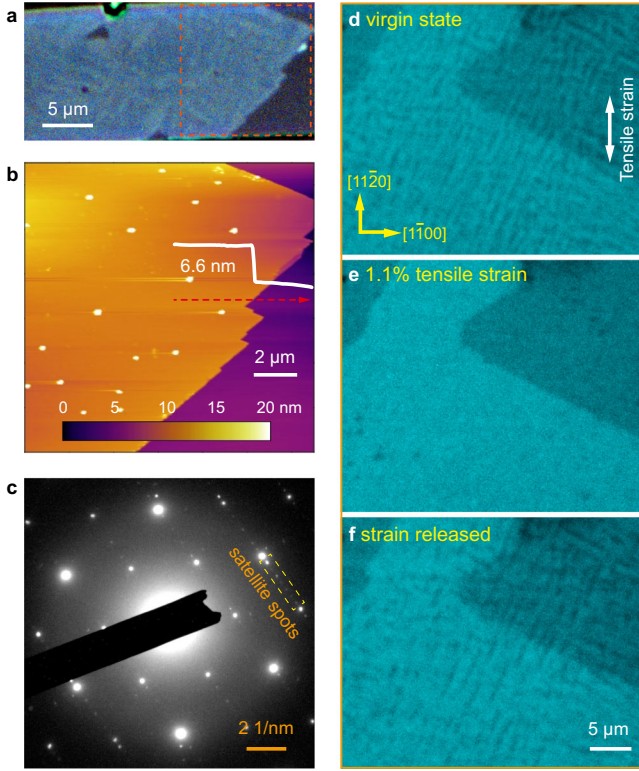

**Fig. 5 2D ferroelasticity in ultrathin $\beta'$-In$_2$Se$_3$ flakes grown by CVD. a** Polarized-light image showing the presence of domain structure in an as-grown ultrathin flake. **b** AFM image of the area highlighted by the orange box in **a**. The inset line profile indicates a thickness of ~6.6 nm. **c** SAED pattern of an ultrathin $\beta'$-In$_2$Se$_3$ flake, revealing the preservation of superstructure indicated by the satellite diffraction. **d–f** Polarized-light images showing ferroelastic domain switching in an ultrathin $\beta'$-In$_2$Se$_3$ flake upon applying the tensile strain.

inclined DWs are again intrinsic to 3R $\beta'$-In$_2$Se$_3$ and do not exist in shear-strain-free 2H $\beta'$-In$_2$Se$_3$ (Supplementary Fig. 11).

## Discussion

In summary, we have demonstrated experimentally 2D ferroelasticity in both exfoliated and CVD-grown $\beta'$-In$_2$Se$_3$ down to few-layer thickness. The spontaneous strain comes from in-plane anti-ferroelectric distortion, which is purely 2D in 2H $\beta'$-In$_2$Se$_3$ in contrast to the conventional 3D ferroelastics. Ferroelastic domain switching between the three variants can be achieved under $\leq$0.5% external strain, offering an easy approach to tailor the antiferro-electric polar structure as well as DW patterns through mechanical stimuli. The detailed domain switching mechanism through both DW propagation and domain nucleation has been unraveled. Furthermore, the effects of 3D stacking on such 2D ferroelasticity, such as surface wrinkles and inclined DWs, have also been revealed explicitly in 3R $\beta'$-In$_2$Se$_3$. We note that during the review process of this work, 2D ferroelasticity is also reported in layered perovskites[32], but only limited to crystals ~10 micron thick in contrast to the 2D flakes down to few-layer thickness investigated in our work. The demonstrated 2D ferroelasticity in both works further indicates its wide presence in vdW materials with anisotropic lattice strain, such as the 1T′ TMDs with Peierls distortion[1,17,28,52,56] and 2D ferro-electrics such as the SnTe family[13,57,58]. Considering a variety of emerging functionalities possessed by these vdW materials, ranging from quantum spin Hall effect to in-plane ferroelectricity[4], it ren-ders exciting potential to tune these 2D functionalities through strain or DW engineering utilizing the ferroelastic behavior. In addition, the thermoelastic transition in $\beta'$-In$_2$Se$_3$ may represent a promising system possessing 2D shape-memory effect, which is worth further investigation.

## Methods

**Materials and sample preparation**. $\beta'$- and $\alpha$-In$_2$Se$_3$ crystals were purchased from HQ Graphene and Alfa Aesar. The $\beta'$-In$_2$Se$_3$ thin flakes were obtained by either direct exfoliation from bulk $\beta'$-In$_2$Se$_3$ or annealing the exfoliated $\alpha$-In$_2$Se$_3$ thin

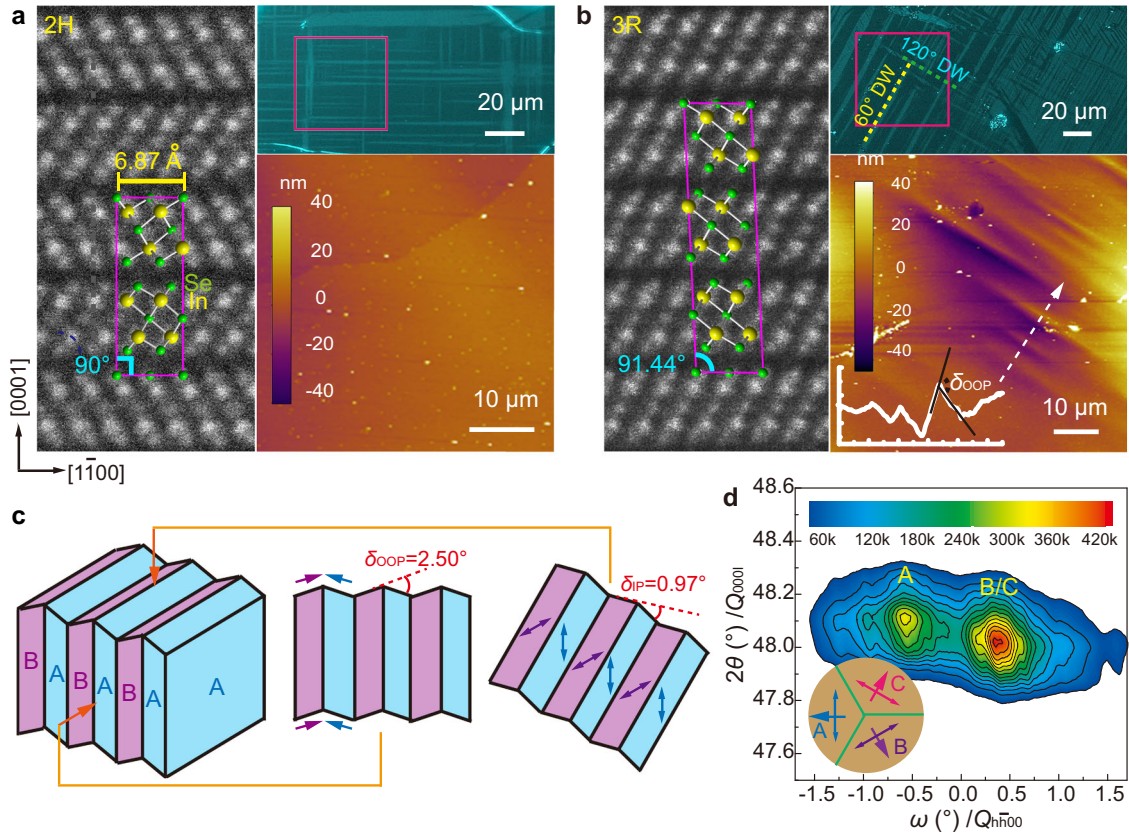

**Fig. 6 Surface wrinkles across 120° DWs owing to the 3D stacking in 3R β'-In₂Se₃. a, b** Atomic-resolution cross-sectional STEM images with the structure models overlaid, and polarized-light images showing multidomain patterns, with AFM topographic images taken from the highlighted regions, for 2H (**a**) and 3R (**b**) β'-In₂Se₃ crystals. 3R β'-In₂Se₃ displays a majority of 60° DWs and minor 120° DWs, with surfaces wrinkles across 120° DWs as revealed by the surface profile in the lower-left inset. In contrast, 60° DWs in the same region have the flat surface. **c** Schematic drawing of the 120° domain pattern in 3R β'-In₂Se₃, showing surface wrinkles across 120° DWs with a 2.50° out-of-plane (OOP) mosaic tilt. The associated in-plane (IP) tilt is ~0.97° (see Supplementary Fig. 8). The IP strain is indicated by double-headed arrows and the shear strain by single-headed arrows. **d** Reciprocal space map with 2θ/ω scan around (0 0 0 15) reflections. The two reflections indicate the OOP mosaic tilt due to the shear strain illustrated in **c**. The directions of the shear strain with respect to the IP strain are indicated in the lower left inset by single-headed arrows.

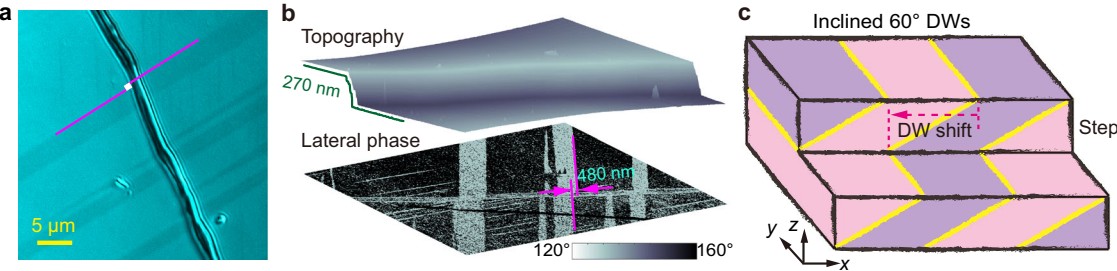

**Fig. 7 Inclined 60° DWs owing to the 3D stacking in 3R β'-In₂Se₃. a** Polarized-light image showing the domain wall shift at the top surface across a step edge. **b** AFM topography and lateral PFM phase images of the area in **a**. The large in-plane piezoresponse in 3R β'-In₂Se₃ gives domain contrast in the lateral PFM phase image, which is not visible in the corresponding AFM image. The measured height change at the step edge is ~270 nm and the wall shift at the top surface is ~480 nm. **c** Schematic drawing illustrating the shift of the intersection lines between the inclined 60° DWs and the top surface, across a step edge in 3R β'-In₂Se₃.

flakes up to 300 °C. The thickness of the exfoliated sample flakes is usually 25−150 nm. Ultrathin β'-In₂Se₃ flakes were grown by the chemical vapor deposition (CVD) method. Specifically, Se and In₂O₃ powders were used as the raw materials and they were placed at separate quartz boats in a dual temperature zone tube furnace. The Se source was placed in the upstream under a 10% H₂/Ar mix carrier gas at a flowing rate of ~20 sccm. The Se and In₂O₃ powders were first heated to 270 and 680 °C respectively and then dwelled for about 30 min, during which thin In₂Se₃ flakes were deposited on the mica substrates placed above the In₂O₃ powders. After the deposition process, the tube furnace was naturally cooled down to room temperature.

We discovered that upon naturally cooling the high-temperature β phase (220−500 °C) with thickness below 400 nm to room temperature, β'-In₂Se₃ always

results and keeps stable. Thus besides direct exfoliation from bulk β'-In₂Se₃, we can also consistently obtain the β' phase by naturally cooling the CVD-grown In₂Se₃, or by annealing thin α-In₂Se₃ flakes at ~300 °C. In₂Se₃ powders for XRD test were ground from a bulk crystal. Specimens for TEM characterization were further thinned by sonication in ethanol and then dispersed onto holey carbon-coated Cu grids. Cross-sectional samples were prepared using a focused ion beam system (JEOL JIB-4500) and then lifted onto holey carbon-coated Cu grids.

**Domain imaging by polarized-light microscopy and piezoresponse force microscopy.** Multidomain patterns were imaged using a polarized-light micro-scope in transmission or reflection mode and recorded with a Leica CCD. The

substrates for mechanically exfoliated sample flakes were $SiO_2@Si$ and for CVD grown flakes were mica. The domain orientations were determined by analyzing the angular dependence of the domain intensity as well as the domain wall configurations. The corresponding surface structure and piezoresponse across the domain walls were also mapped using a commercial atomic force microscope (AFM, Asylum Research MFP-3D) with a conductive tip of ~2.8 N/m spring constant.

**Structure analysis using TEM/STEM and XRD.** The structure of 2H and 3R $\beta'$-$In_2Se_3$ was identified by conventional TEM and electron diffraction conducted on a JEOL JEM-2100F TEM operated at 200 kV. In situ TEM heating was carried out on E-chips using a Protochip Fusion holder. Atomic-resolution STEM imaging was performed on a JEOL JEM ARM 200CF microscope equipped with a cold field emission gun and an ASCOR fifth-order probe corrector. It was operated at 200 kV with a 26 mrad convergence semi-angle, and the collection semi-angle for ADF signal was 68−200 mrad. Reciprocal space mapping (RSM) and powder X-ray diffraction (PXRD) were performed on a Rigaku X-ray diffractometer (SmartLab 9 kW). Lattice parameters were calculated by Rietveld refinement using least square weighting model based on the PXRD data.

**Ferroelastic domain switching.** Controllable uniaxial tensile strain was applied to exfoliated (thickness of 50−150 nm) or CVD-grown (thickness of ~15 nm) $\beta'$-$In_2Se_3$ flakes by two-point bending the PET substrates[54]. The magnitude of strain ($\varepsilon$) was derived from the radius of curvature ($R$) and substrate thickness ($h$) with the formula $\varepsilon = h/2R$. To prevent slippage between $\beta'$-$In_2Se_3$ flakes and substrates, each tested flake was encapsulated by a thin PMMA layer[59] and then attached to the PET substrates using gauge bonding agent (Kisling Ergo 5400). The strain was further checked by strain gauges to ensure the effective strain transfer from substrates to the bonded flakes (see Supplementary Fig. 2). The domain switching behavior was in situ captured during straining under the polarized-light microscope.

## Data availability
The data that support this study are available from the corresponding author upon reasonable request.

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

## Acknowledgements

Y.Z. is thankful for the financial support from the Research Grants Council of Hong Kong (No. 15305718) and the Hong Kong Polytechnic University grant (No. ZVGH). J.H. is thankful for the General Research Fund from the Research Grants Council of Hong Kong (No. PolyU 153033/17P). X.L. is thankful for the support from the National Natural Science Foundation of China (Nos. 11804286 and 11832019), the Fundamental Research Funds for the Central Universities, and the Natural Science Foundation of Guangdong Province (No. 2021B1515020021). D.L. is thankful for the financial support from the Research Grants Council of Hong Kong (Project No. 15303718). Technical support of the high-resolution electron microscopy facility at MCPF of HKUST is also acknowledged.

## Author contributions

Y.Z., C.Z. and C.X. conceived the project. Y.Z. led the project. C.X. and S.Y. prepared the samples and conducted the ferroelastic switching. J.M. and J.H. grew the ultrathin flakes. C.X. and X.G. performed the TEM/STEM observation. X.L. and Y.C. carried out the structure calculations. T.L. and D.L. helped with the linear dichroism measurement. C.C. assisted with the AFM and PFM test. C.X. and Y.Z. analyzed the experimental data and wrote the manuscript. All authors discussed the results and commented on the manuscript.

## Competing interests

The authors declare no competing interests.
