## [Peer Review File · Nature Communications]

REVIEWER COMMENTS

Reviewer #1 (Remarks to the Author):

In this work, the authors report the study on the ferroelasticity in α' -In₂Se₃. The finding is the identification of 2D spontaneous strain in in-plane antiferroelectric α' -In₂Se₃ via the multiple characterization with STEM, in situ XRD, and electron diffraction techniques. Overall, the data is beautiful and the results are interesting. However, there are several issues need to be addressed before its further consideration.

1. Since the authors aim to demonstrate 2D ferroelasticity, they need to be aware of a recent work on 2D ferroelasticity in layered perovskites (Nat. Commun. 12, 1332 (2021)).
2. To some extent the manuscript is lengthy. The authors have written quite a few paragraphs and presented several figures in the maintext to describe the atomic structure and the domain structure of α' -In₂Se₃. The authors should have noticed that several published papers have already clearly reported the same lattice structure, including the phase transition study in single layer α' -In₂Se₃ (ACS Nano 13, 8004-8011 (2019)) and the author's previous findings on in-plane ferroelectricity (Sci. Adv. 4, 7720 (2018)) and antiferroelectricity (PRL 125, 047601 (2020)). Some results in this manuscript, i.e. the Fig. 1b, Fig. 1d, and Fig. 2b, are similar with those in the PRL paper. The novelty lies in the data processing, which yields the non-zero in-plane strain distribution. I believe these figures are important to demonstrate the claimed ferroelasticity, but I would like to suggest the authors present these data in a brief way.
3. The authors show the evidence of ferroelasticity in α' -In₂Se₃ via the study on the domain dynamics under varied external strain. However, when applying the uniaxial tensile strain, the authors use the bending technique via a flexible substrate. And the strain is estimated through the curvature effect. Here, though the α' -In₂Se₃ is fixed by encapsulating PMMA, the questions are how the strain is transferred from the substrate and how precise the method is. Additionally, using bending, the flexoferroelectric effect cannot be ignored especially for ultrathin vdW ferroelectrics. For ferroelectrics, the mechanical bending can result in electric polarization flipping which leads to similar domain switching. Since α' -In₂Se₃ is a kind of vdW ferroelectrics, to exclude flexoferroelectric effect, the authors needs to present smoking gun for the claimed ferroelasticity in α' -In₂Se₃. I would like to suggest the authors to elaborate on this point.
4. The CVD sample could be as thin as 6.6 nm. What is the thickness for the mechanical exfoliated samples?

Reviewer #2 (Remarks to the Author):

The manuscript by Xu et al. reports a comprehensive study on the discovery of spontaneous ferroelasticity in β' -In₂Se₃ by scanning transmission electron microscopy (STEM), X-ray diffraction (XRD), polarized light microscopy (PLM), and piezo-force microscopy (PFM) techniques. It should be noted that they have reported similar STEM and PLM data in a prior publication (Ref. 18). The new aspects in this paper is the PLM observation of domain switching by the application of uni-axial tensile strain, which unequivocally demonstrates the ferroelasticity. A careful and convincing analysis is carried out to understand the various domain walls. Given the sufficient novelty in this work and the current interest in ferroic properties of 2D materials, I believe the paper can be published in Nature Communications if the authors can address the following issues.

- 1) It is well documented (and noted by the authors) that In₂Se₃ has many stable or metastable phases under the ambient conditions. The authors should provide thorough details on how they obtain the β' -In₂Se₃ crystal (especially by the CVD method) such that other people may reproduce their

results. Also, is there an understanding on why they can consistently get the β' rather than α phase? If so, that should be included in Methods or Supplementary Information.

2) Following the previous comment, it is commonly known that Raman spectroscopy is a reliable method to identify different phases of In_2Se_3 . I would encourage that the authors perform and show Raman data on their samples.

3) The discussion of PFM data in Fig. 7 is rather brief. If the samples are indeed ferroelastic rather than ferroelectric, why would they see PFM (electromechanical coupling) contrast at all? A simple modeling or qualitative picture may help with the discussion of PFM results.

Reviewer #3 (Remarks to the Author):

In this paper, The authors Xu et al. claim that they present the first experimental demonstration of 2D ferroelasticity in both exfoliated and CVD-grown β' - In_2Se_3 down to few-layer thickness. The results are interesting and novelty. Before making the final decision whether it can be accepted in Nature Commun., I have some comments as follows.

1) How many layers of the sample for the β' - In_2Se_3 ? As is well known, van der Waals interaction exists in this material. How to rule out the influence of this effect?

2) What is the substrate and how to rule out the influence of substrate here?

3) Could the authors show the strain-stress hysteresis curve for 2D β' - In_2Se_3 ? It may be direct evidence.

4) What is the elastic range of the material before it is broken?

Response to Reviewers

We have provided two copies of the revised manuscript, one with and one without the changes marked-up. Our detailed response to referees' comments follows. (Note: referees' comments are in blue; changes made to the text in the manuscript are copied or referenced here in red.)

Reviewer #1 (Remarks to the Author):

In this work, the authors report the study on the ferroelasticity in β' -In₂Se₃. The finding is the identification of 2D spontaneous strain in in-plane antiferroelectric β' -In₂Se₃ via the multiple characterization with STEM, in situ XRD, and electron diffraction techniques. Overall, the data is beautiful and the results are interesting. However, there are several issues need to be addressed before its further consideration.

1. Since the authors aim to demonstrate 2D ferroelasticity, they need to be aware of a recent work on 2D ferroelasticity in layered perovskites (Nat. Commun. 12, 1332 (2021)).

RESPONSE: We thank the reviewer for the positive assessment and helpful suggestion. We did not notice this paper as it was online only after the submission of our paper. But it is indeed very relevant and we have added it into our revised manuscript as Ref. 32, and have also added a few descriptions in the main text as copied below. In the revised manuscript, we also tried to point out our unique contribution to 2D ferroelasticity – the demonstration on 2D flakes few-layer thick, to distinguish from the above work on the layered perovskites.

The end of first paragraph: However, despite several proposed 2D ferroelastic behavior²⁸⁻³¹ and one very recent demonstration on micron-thick layered perovskites³², 2D ferroelasticity evidenced by the mechanical switch of spontaneous lattice strain in ultrathin vdW materials has not yet been validated experimentally⁴.

The last paragraph: We note that during the review process of this work, 2D ferroelasticity is also reported in layered perovskites³², but only limited to crystals ~10 micron thick in contrast to the 2D flakes down to few-layer thickness investigated in our work.

2. To some extent the manuscript is lengthy. The authors have written quite a few paragraphs and presented several figures in the maintext to describe the atomic structure and the domain structure of β' -In₂Se₃. The authors should have noticed that several published papers have already clearly reported the same lattice structure, including the phase transition study in single layer β' -In₂Se₃ (ACS Nano 13, 8004-8011 (2019)) and the author's previous findings on in-plane ferroelectricity (Sci. Adv. 4, 7720 (2018)) and antiferroelectricity (PRL 125, 047601 (2020)). Some results in this manuscript, i.e. the Fig. 1b, Fig. 1d, and Fig. 2b, are similar with those in the PRL paper. The novelty lies in the data processing, which yields the non-zero in-plane strain distribution. I believe these figures are important to demonstrate the claimed ferroelasticity, but I would like to suggest the authors present these data in a brief way.

RESPONSE: We thank the reviewer for pointing this out. We have wished to make this paper self-standing, which unavoidably leads to some extent of repeat. We agree that there may be too much repeating, and have trimmed down the corresponding text according to the reviewer's suggestion. **Please see the marked-up version to see the changes made.**

Also we wish to point out that even though the outlook may be similar, Fig. 1d is actually for *in situ* XRD instead of electron diffraction in our previous PRL paper (PRL 125, 047601 (2020)). Indeed the two approaches yield consistent trends, but XRD has been considered as a much more reliable and quantitative technique compared to electron diffraction. And here we also included both *in situ* heating and cooling to verify the reversibility of the phase transformation. So we keep Fig. 1d but also shorten the text describing it.

3. The authors show the evidence of ferroelasticity in β' - In_2Se_3 via the study on the domain dynamics under varied external strain. However, when applying the uniaxial tensile strain, the authors use the bending technique via a flexible substrate. And the strain is estimated through the curvature effect. Here, though the β' - In_2Se_3 is fixed by encapsulating PMMA, the questions are how the strain is transferred from the substrate and how precise the method is.

RESPONSE: We appreciate the reviewer's concern. Our substrate bending test is a demonstrated strategy that has been widely used to measure the mechanical properties of vdW materials (e.g., Nano Lett. 17, 6097 (2017); J. Phys.: Condens. Matter 27, 313201 (2017); Nat. Commun. 11, 1151, (2020)); and also the recent work on 2D ferroelasticity in layered perovskites (Nat. Commun. 12, 1332 (2021) pointed out by the reviewer). Especially with polymer encapsulating, vdW flakes should be rigidly clamped onto the substrates without slippage (e.g., Nat. Commun. 11, 1151, (2020)). To further exclude the potential slippage, we have coated the substrates with a small drop of gauge bonding agent (Kisling Ergo 5400) to ensure the full clamping of vdW flakes. Thus the flakes should be under uniaxial strain equal to the surface strain of the substrates. The experimental method has been briefly described in **Result - Ferroelastic domain switching by external strain** session, the 1st paragraph: "By gluing it onto a flexible substrate and bending it upward, we then stretch the flake with external tensile strain ..."

To double check the clamping effect, we have used the same bonding agent to attach strain gauges on the substrates and calibrated the strain directly. As shown below, the strain values derived from both the strain gauges and the curvature measurement show excellent consistency and thus confirm the complete strain transfer from substrates to bonded materials. We believe that this new calibration validates the strain values presented in our manuscript. **The result has been added in Supplementary Information as Supplementary Fig. 2 and also described in Methods: Ferroelastic domain switching.**

Supplementary Figure 2. Strain calibration using strain gauge. a Radii of curvature (R) measurement in the two-point bending experiment. **b** Strain values obtained from the substrate radii of curvature and the two strain gauges attached on the substrates. The lower inset in **b** shows the geometry of the two-point bending setup. The strain in each step is controlled by setting the relative displacement Δx of the two terminals.

Change made in Methods: Ferroelastic domain switching

... To prevent slippage between β' - In_2Se_3 flakes and substrates, each tested flake was encapsulated by a thin PMMA layer⁵⁹ and then attached to the PET substrates using gauge bonding agent (Kisling Ergo 5400). The strain was further checked by strain gauges to ensure the effective strain transfer from substrates to the bonded flakes (see Supplementary Fig. 2). ...

Additionally, using bending, the flexoferroelectric effect cannot be ignored especially for ultrathin vdW ferroelectrics. For ferroelectrics, the mechanical bending can result in electric polarization flipping which leads to similar domain switching. Since β' - In_2Se_3 is a kind of vdW ferroelectrics, to exclude flexoferroelectric effect, the authors need to present smoking gun for the claimed ferroelasticity in β' - In_2Se_3 . I would like to suggest the authors to elaborate on this point.

RESPONSE: We agree with the reviewer that the flexoferroelectric effect may cause flipping of polarization and domain switching due to strain induced ferroelectric transition. To exclude this effect, we have performed *in situ* Raman spectroscopy and second harmonic generation (SHG) on the strained β' - In_2Se_3 . As shown below, Raman spectra confirm the persistence of the β' structure which is antiferroelectric in nature. The absence of SHG signal (in contrast to the strong SHG signal from ferroelectric α - In_2Se_3) further rule out the possibility of ferroelectric

transition during the domain switching process. This new result has been added in **Supplementary Information as Supplementary Figs. 3b and 3c** and also mentioned in main text.

Supplementary Figures 3b and 3c. **b** *In situ* Raman spectroscopy and **c** SHG from tensile-strained β' - In_2Se_3 . An α - In_2Se_3 flake is taken for reference in the SHG measurement as there is no SHG response in β' - In_2Se_3 . Raman spectroscopy was carried out on a Witec confocal microscopy with a 532 nm exciting laser. SHG was collected with a monochromator (Princeton SpectraPro 2750 integrated with a ProEM EMCCD camera with a spectral resolution less than 0.1 nm) under a Ti:sapphire femtosecond laser source (Coherent Libra) centered at 800 nm.

Change made in main text (page 7, 2nd paragraph): We have explicitly excluded the potential phase change or flexoferroelectric effect during strain-induced domain switching, using *in situ* Raman spectroscopy and second harmonic generation (see Supplementary Fig. 3).

4. The CVD sample could be as thin as 6.6 nm. What is the thickness for the mechanical exfoliated samples?

RESPONSE: The thickness of the mechanically exfoliated β' - In_2Se_3 flakes can be as thin as 25 nm (see the lower-right corner of Fig. R1D below). However, thin flakes can easily wrinkle after exfoliation, and to better observe domain contrast in a large flat region, we thus picked relatively thick flakes ~50-150 nm for domain switching. The thickness measurement by AFM and the corresponding domain patterns are present below. This information has been added in **Methods - Materials and sample preparation** as: The thickness of the exfoliated sample flakes is usually 25–150 nm, and in **Methods - Ferroelastic domain switching** as: Controllable uniaxial tensile strain was applied to exfoliated (thickness of 50–150nm) or CVD-grown (thickness of ~15 nm) β' - In_2Se_3 flakes by two-point bending the PET substrates.

Figure R1. Optical images and AFM topography of mechanically exfoliated β' - In_2Se_3 samples. An overview of the flakes is shown in the top left. The AFM topography of regions A–F is shown in the bottom left, with the average thicknesses labelled (Scale bar: 5 μm). The right column presents the polarized light images showing domain patterns in regions B, D and E with flake thickness varies from 25 nm to 150 nm.

Reviewer #2 (Remarks to the Author):

The manuscript by Xu et al. reports a comprehensive study on the discovery of spontaneous ferroelasticity in β' -In₂Se₃ by scanning transmission electron microscopy (STEM), X-ray diffraction (XRD), polarized light microscopy (PLM), and piezo-force microscopy (PFM) techniques. It should be noted that they have reported similar STEM and PLM data in a prior publication (Ref. 18). The new aspects in this paper are the PLM observation of domain switching by the application of uni-axial tensile strain, which unequivocally demonstrates the ferroelasticity. A careful and convincing analysis is carried out to understand the various domain walls. Given the sufficient novelty in this work and the current interest in ferroic properties of 2D materials, I believe the paper can be published in Nature Communications if the authors can address the following issues.

1) It is well documented (and noted by the authors) that In₂Se₃ has many stable or metastable phases under the ambient conditions. The authors should provide thorough details on how they obtain the β' -In₂Se₃ crystal (especially by the CVD method) such that other people may reproduce their results.

RESPONSE: We are thankful for the reviewer's positive assessment and helpful comments. We have added the ways to achieve β' -In₂Se₃ phase in **Methods - Materials and sample preparation** in the revised manuscript:

The β' -In₂Se₃ thin flakes were obtained by either direct exfoliation from bulk β' -In₂Se₃ or annealing the exfoliated α -In₂Se₃ thin flakes up to 300 °C. The thickness of the exfoliated sample flakes is usually 25–150 nm. Ultrathin β' -In₂Se₃ flakes were grown by the chemical vapor deposition (CVD) method. Specifically, Se and In₂O₃ powders were used as the raw materials and they were placed at separate quartz boats in a dual temperature zone tube furnace. The Se source was placed in the upstream under a 10% H₂/Ar mix carrier gas at a flowing rate of ~20 sccm. The Se and In₂O₃ powders were first heated to 270 and 680 °C respectively and then dwell for about 30 min, during which thin In₂Se₃ flakes were deposited on the mica substrates placed above the In₂O₃ powders. After the deposition process, the tube furnace was naturally cooled down to room temperature.

Also, is there an understanding on why they can consistently get the β' rather than α phase? If so, that should be included in Methods or Supplementary Information.

RESPONSE: We have added our understanding of forming the β' rather than α phase in **Methods - Materials and sample preparation** in the revised manuscript:

We discovered that upon naturally cooling the high-temperature β phase (220–500 °C) with thickness below 400 nm to room temperature, β' -In₂Se₃ always results and keeps stable. Thus besides direct exfoliation from bulk β' -In₂Se₃, we can also consistently obtain the β' phase by naturally cooling the CVD-grown In₂Se₃, or by annealing thin α -In₂Se₃ flakes at ~300 °C.

2) Following the previous comment, it is commonly known that Raman spectroscopy is a reliable method to identify different phases of In_2Se_3 . I would encourage that the authors perform and show Raman data on their samples.

RESPONSE: We appreciate the reviewer's constructive suggestion. We have carried out Raman spectroscopy on the 2H/3R stacking α -/ β' - In_2Se_3 with 532 nm excitation laser and it indeed can distinguish different phases as shown below. This new result has been added in **Supplementary Information as Supplementary Fig. 3a**.

Supplementary Figure 3a. a Raman spectra for α -/ β' - In_2Se_3 with 2H/3R stacking with the low-frequency Raman shifts labelled in each spectrum. Raman peaks near 86, 102, 178, 185, and 193 cm^{-1} are observed in 2H/3R α - In_2Se_3 , and near 60, 106, 173, and 202 cm^{-1} are observed in 2H/3R β' - In_2Se_3 flakes, consistent with previous studies [Nano Lett. 13, 3501-3505, (2013); Chem. Mater. 31, 10143-10149, (2019)]. These peaks are merely related to the intralayer structure and independent on the stacking order, and thus can be used to distinguish α - and β' - In_2Se_3 . In addition to the distinct peak positions, β' - In_2Se_3 also shows broader Raman peaks than α - In_2Se_3 due to the strong vibrational anharmonicity [J. Phys. Chem. C 122, 22849, (2018)]. In the low-frequency range from 10 to 50 cm^{-1} , there are peaks at 18 and 26 cm^{-1} for 2H and 3R α - In_2Se_3 respectively, and peaks at 14 and 24 cm^{-1} for 2H and 24 and 31 cm^{-1} for 3R β' - In_2Se_3 .

These Raman peaks are presumably related to the interlayer vibrational modes, and can thus be used to distinguish the interlayer stacking order.

3) The discussion of PFM data in Fig. 7 is rather brief. If the samples are indeed ferroelastic rather than ferroelectric, why would they see PFM (electromechanical coupling) contrast at all? A simple modeling or qualitative picture may help with the discussion of PFM results.

RESPONSE: We attribute the observed PFM response from ferroelastic β' - In_2Se_3 (antiferroelectric) to a second-order electrostrictive effect rather than ferroelectricity. Following the reviewer's suggestion, we have added a new **Supplementary Note 5** and also copied below:

In typical ferroelectrics, the first-order electrostrictive effect (piezoelectricity) is strong and dominates the PFM response. As illustrated in Supplementary Figs. 9a and 9b, under a modulated cantilever voltage $V_{ac} = V_0 \cos \omega t$, the out-of-plane (OOP) and in-plane (IP) piezoresponses can be described as

$$\Delta z = d_{33}V_{ac},$$

$$\Delta x = d_{15}V_{ac},$$

where d_{33} and d_{15} are the longitudinal and shear piezocoefficients, respectively. As demonstrated by Sader [J. Appl. Phys. 84, 64 (1998)], in the vicinity of a resonance with small damping (quality factor $Q > 10$), the amplitude and phase frequency responses can be described using the harmonic oscillator model as

$$A(\omega) = \frac{A_{max}\omega_0^2/Q}{\sqrt{(\omega_0^2 - \omega^2)^2 + (\omega_0\omega/Q)^2}}$$

$$\tan \varphi(\omega) = \frac{\omega_0\omega}{Q(\omega_0^2 - \omega^2)}.$$

Considering two oppositely oriented ferroelectric domains with IP polarization, for instance, the PFM responses are opposite in sign

$$\Delta x_+(\omega) = d_{15}V_{ac} = d_{15}V_0 \cos \omega t,$$

$$\Delta x_-(\omega) = -d_{15}V_{ac} = d_{15}V_0 \cos(\omega t + \pi).$$

Thus in ferroelectrics, according to the above equations, there is a phase jump of 180° across the resonant point and a phase difference of 180° off the resonance between oppositely oriented domains [J. Phys. D: Appl. Phys. 44, 464003 (2011)].

On the other hand, non-ferroelectric dielectrics can also show electromechanical response through the second-order electrostrictive effect, which is quadratic in the applied voltage [Appl. Phys. Lett. 104, 242907 (2014); ACS nano 5, 9104 (2011)], as shown in Supplementary Figs. 9c and 9d. Thus the OOP and IP PFM response are

$$\Delta z = Q_{3333}(\varepsilon_{33}V_{ac})^2 = Q_{3333}V_{ac}^2\cos^2\omega t = Q_{3333}V_{ac}^2(\cos 2\omega t + 1)/2$$

$$\Delta x = Q_{1333}(\varepsilon_{33}V_{ac})^2 = Q_{1333}V_{ac}^2\cos^2\omega t = Q_{1333}V_{ac}^2(\cos 2\omega t + 1)/2.$$

These equations suggest a second-harmonic response with the driving force resulted from the second-order electrostrictive effect, which is significantly different from the piezoelectricity. Considering oppositely oriented domains with IP shear strain, the second-harmonic responses are

$$\Delta x_+(\omega) = Q_{1333}V_{ac}^2(\cos 2\omega t + 1)/2$$

$$\Delta x_-(\omega) = -Q_{1333}V_{ac}^2(\cos 2\omega t + 1)/2.$$

Thus as long as Q_{1333} is non-zero (non-zero IP shear strain), the domains can still be imaged. But the phase change from this nonlinear response will be $\sim 90^\circ$ for ω instead of 180° . On the other hand, the current PFM instrument is not designed to work on such second-order electrostrictive effect, thus the measured the phase difference may deviate from 90° , but definitely much less than 180° .

To further illustrate the non-ferroelectric origin of the PFM contrast, Supplementary Fig. 10 compares the lateral PFM response of the 3R β' - In_2Se_3 flake with the ferroelectric lead zirconate titanate (PZT) thin film. As expected, the amplitude/phase frequency responses as well as the lateral PFM imaging between the two materials behave differently. The PZT thin film exhibits typical ferroelectric behavior – a phase change of 180° across the resonant point and a phase difference of 180° between domains, while the 3R β' - In_2Se_3 flake shows smaller phase difference that are consistent with the above second-order electrostrictive effect for non-ferroelectrics. Furthermore, for 2H β' - In_2Se_3 , because of the zero in-plane

electromechanical response ($Q_{1333} = 0$) of the orthorhombic lattice (Supplementary Fig. 9c), consequently there's negligible contrast over the ferroelastic domains, as shown in Supplementary Fig. 11, providing another evidence that the domains in β' - In_2Se_3 are not ferroelectric. Therefore, we conclude that the domain contrast in 3R β' - In_2Se_3 arises from the second-order electrostrictive effect rather than ferroelectricity.

Supplementary Figure 9. PFM measurements on typical ferroelectrics and non-ferroelectrics. a, b Piezoelectric response characterized by PFM with OOP polarization (a) and IP polarization (b). **c, d** Piezoelectric response characterized by PFM in non-ferroelectrics with orthorhombic lattice (c) and monoclinic lattice (d). See Supplementary Note 5 for detailed discussion.

Supplementary Figure 10. Lateral PFM on a ferroelectric lead zirconate titanate (PZT) thin film and a non-ferroelectric 3R β' -In₂Se₃ flake. **a, d** In-plane amplitude and phase frequency responses of two different domains in a PZT thin film (a) and a 3R β' -In₂Se₃ flake (d) across the resonance. **b, c** Amplitude and **c, f** phase mapping of the PZT thin film (b, c) and 3R β' -In₂Se₃ flake (e, f) under a driven frequency slightly above the resonance. The less than 180° phase difference both across the resonance and between the domains in 3R β' -In₂Se₃ indicates its non-ferroelectric nature in contrast to PZT. See Supplementary Note 5 for detailed discussion.

Supplementary Figure 11. DWs across a step edge and the corresponding lateral PFM images in 2H β' - In_2Se_3 . **a** Polarized light image showing no horizontal shift of DWs across the step edge, reflecting the vertical nature in contrast to the inclined DWs in 3R β' - In_2Se_3 . **b-d** AFM topography (b), lateral PFM amplitude (c) and phase (d) images of the area highlighted by the black box in **a**. No domain contrast is observed in lateral PFM images, further confirming the non-ferroelectric nature of these ferroelastic domains.

Reviewer #3 (Remarks to the Author):

In this paper, the authors Xu et al. claim that they present the first experimental demonstration of 2D ferroelasticity in both exfoliated and CVD-grown β' -In₂Se₃ down to few-layer thickness. The results are interesting and novelty. Before making the final decision whether it can be accepted in Nature Commun., I have some comments as follows.

1) How many layers of the sample for the β' -In₂Se₃? As is well known, van der Waals interaction exists in this material. How to rule out the influence of this effect?

RESPONSE: We are thankful for the reviewer's helpful comments. For mechanically exfoliated β' -In₂Se₃ flakes, they are usually 25–150-layer thick as measured by AFM shown in Fig. R1 below (~0.9 nm/layer). The CVD-grown flakes can be down to 6 layers (~5.4 nm) with detectable domain contrast. DFT calculations indicate stable 2D ferroelastic strain coupled to antiferroelectricity down to the single layer limit [PRL 125, 047601 (2020)], which is consistent with our demonstration of 2D in-plane ferroelasticity in 2H β' -In₂Se₃. On the other hand, vdW interaction can indeed affect 2D ferroelasticity, as evidenced by the distinct ferroelastic behavior in 3R β' -In₂Se₃ (Figs. 6 and 7) which has been attributed to its different stacking order. Such a stacking order effect is presumably a result of vdW interaction, since it is the only interlayer interaction present. Overall, we have demonstrated ferroelasticity in both 2H and 3R β' -In₂Se₃ despite of their different vdW interactions.

This information has been added in **Methods - Materials and sample preparation** as: The thickness of the exfoliated sample flakes is usually 25–150 nm, and in **Methods - Ferroelastic domain switching** as: Controllable uniaxial tensile strain was applied to **exfoliated (thickness of 50–150nm) or CVD-grown (thickness of ~15nm)** β' -In₂Se₃ flakes by two-point bending the PET substrates.

Figure R1. Optical images and AFM topography of mechanically exfoliated β' - In_2Se_3 samples. An overview of the flakes is shown in the top left. The AFM topography of regions A–F is shown in the bottom left, with the average thicknesses labelled (Scale bar: 5 μm). The right column presents the polarized light images showing domain patterns in regions B, D and E with flake thickness varies from 25 nm to 150 nm.

2) What is the substrate and how to rule out the influence of substrate here?

RESPONSE: We appreciate the reviewer’s concern. The substrates for mechanical exfoliated sample flakes are SiO_2/Si and for CVD grown flakes are mica, and in both systems the ferroelastic domains are clearly seen. These domains persist when transferred to PET substrates for ferroelastic domain switching, and to holey-carbon coated Cu grids for TEM observation. Thus we can conclude that the structure of β' - In_2Se_3 is not affected by substrates. **We have added this information to *Methods - Domain imaging by polarized-light microscopy and piezoresponse force microscopy*: The substrates for mechanically exfoliated sample flakes are SiO_2/Si and for CVD grown flakes are mica.**

Regarding the ferroelastic domain switching on the bendable PET substrates, the substrate acts as a media for transferring strain to the thin flakes, which is a well-established approach for strain engineering the structure and properties of vdW materials and thin films [Nature 514, 470 (2014); Sci. Adv. 3, e1602165, (2017); Nat. Commun. 12, 1332 (2021)]. To our best knowledge, there is no mechanism that the PET substrates may affect the ferroelastic domain switching behavior beyond the role of strain transfer. We have also added Supplementary Figs. 2 and 3 to prove that the tensile strain can be effectively applied to the flakes through the PET substrate and the β' structure doesn't change during this process. This new result has been added in Supplementary Information as Supplementary Figs. 3b and 3c and also mentioned in main text.

Supplementary Figures 3b and 3c. **b** *In situ* Raman spectroscopy and **c** SHG from tensile-strained β' - In_2Se_3 . An α - In_2Se_3 flake is taken for reference in the SHG measurement as there is no SHG response in β' - In_2Se_3 . Raman spectroscopy was carried out on a Witec confocal microscopy with a 532 nm exciting laser. SHG was collected with a monochromator (Princeton SpectraPro 2750 integrated with a ProEM EMCCD camera with a spectral resolution less than 0.1 nm) under a Ti:sapphire femtosecond laser source (Coherent Libra) centered at 800 nm.

Change made in main text (page 7, 2nd paragraph): We have explicitly excluded the potential phase change or flexoferroelectric effect during strain-induced domain switching, using *in situ* Raman spectroscopy and second harmonic generation (see Supplementary Fig. 3).

3) Could the authors show the strain-stress hysteresis curve for 2D β' - In_2Se_3 ? It may be direct evidence.

RESPONSE: We agree with the reviewer that the classical strain-stress hysteresis curve is a direct evidence to ferroelasticity. However, such measurement is extremely difficult to achieve on 2D flakes that are usually < 0.1 mm in size. We have tried the AFM nano-indentation approach to acquire the strain-stress curve in suspended flakes. However, as added in

Supplementary Information as Supplementary Fig. 4 (also copied below on page 19), with the complication of the large pre-strain (which leads to a dome shape for the suspended area under test) and strong adhesive force between the AFM tip and the flake upon unloading, it is impractical to obtain the strain-stress curve associated with ferroelastic-domain switching. Thus we choose to directly observe the domain switching under applied mechanical stress/strain, which is an established method demonstrated by several impactful works [Sci. Adv. **3**, e1602165, (2017); Nat. Commun. **12**, 1332 (2021)]. Using the domain area as an indicator, we can sort of demonstrate the hysteresis behavior as illustrated in Fig. 4h - **we have changed the origin of this curve in the revised manuscript as shown below**, which now looks more consistent with the classical hysteresis curve for ferroelastics.

Figure 4

4) What is the elastic range of the material before it is broken?

RESPONSE: We thank the reviewer for this question. We have conducted AFM nano-indentation test to evaluate the breaking strength and strain of β' -In₂Se₃. **This new result has been added in Supplementary Information as Supplementary Fig. 4 and Note 3** and also mentioned in main text, as copied below:

The AFM nano-indentation is a common method for studying the mechanical properties of 2D materials [Science **321**, 385-388 (2008); ACS Nano **5**, 9703-9709 (2011); Nat. Commun. **8**, 15815, (2017); ACS Nano **12**, 10347-10354, (2018)]. Here we conducted AFM indentation test on a

CVD-grown flake of ~14 nm thick to estimate the breaking strength of vdW β' -In₂Se₃. As shown in Supplementary Fig. 4, the membrane is suspended over a circular hole with radius $r = 750$ nm and deformed in the center by an AFM tip (with a radius $r_{\text{tip}} < 25$ nm).

With $r_{\text{tip}} \ll r$, the classical theory of continuum mechanics can be applied [ACS Nano 5, 9703-9709 (2011)]. The deformation behavior of a suspended 2D flake during indentation can be approximated by the equation [J. Appl. Mech. 72, 203–212 (2005)]:

$$F = \sigma_0^{2D} \pi \delta + E^{2D} \frac{q^3 \delta^3}{r^2}.$$

The first term linear with deflection δ corresponds to the linear pre-stretched membrane regime with σ_0^{2D} as the prestress in the membrane. The second term represents the modified form of the classical Schwerin solution for point loading on the circular suspended thin membrane and is related to the elastic modulus E^{2D} . The dimensionless constant q depends on the Poisson ratio ν as $q = 1/(1.05 - 0.15\nu - 0.16\nu^2)$. The Poisson ratio of $\nu \sim 0.25$ for vdW In₂Se₃ gives $q=0.998$ [Solid State Commun. 325, 114159 (2021)].

By fitting the measured force-deflection data (Supplementary Fig. 4c) using the above formula, the elastic modulus E^{2D} of the β' -In₂Se₃ 2D flake can be extracted to be 1429 ± 178 N/m. With the derived elastic modulus, the maximum stress σ_{max}^{2D} can then be calculated to be 80 ± 4 N/m using the expression for the indentation of a clamped, linearly elastic membrane by a small spherical indenter ($r_{\text{tip}}/r \ll 1$, with $r_{\text{tip}} = 25$ nm and $r = 750$ nm) at its loading limit:

$$\sigma_{\text{max}}^{2D} = \sqrt{\frac{F_{\text{max}} E^{2D}}{4\pi r_{\text{tip}}}}.$$

Considering the flake thickness of ~14 nm, the effective Young's modulus $E^{\text{eff}} = 102 \pm 13$ GPa and breaking strength $\sigma_{\text{max}}^{\text{eff}} = 5.7 \pm 0.3$ GPa can be obtained. These values closely match the predicted ones for monolayer β -In₂Se₃ [Phys. Chem. Chem. Phys. 21, 19234 (2019)], which validates our nano-indentation measurement. A linear approximation of the mechanical relationship between the stress and strain for stiff materials further gives the breaking or yield strain ~5.5% ($\sigma_{\text{max}}^{\text{eff}}/E^{\text{eff}}$) [ACS Nano 12, 10347-10354, (2018)], much larger than the strain required for ferroelastic domain switching.

Supplementary Figure 4. AFM nano-indentation on CVD-grown β' - In_2Se_3 transferred onto holey SiO_2 @ Si substrates. **a** AFM image of a ~ 14 nm thin In_2Se_3 flake suspended over a $1.5 \mu\text{m}$ hole for the indentation experiment. The profile in blue across the holey region (red dashed line) suggests pre-strain of the suspended area. **b** Fracture of the thin In_2Se_3 flake after large-force loading. **c** The typical loading curves for a β' - In_2Se_3 flake and the least-squares fit. The fracture point is marked by the symbol *. The inset shows the schematic of the indentation experiment on the suspended In_2Se_3 flake.

Change made in main text (page 7, 2nd paragraph): The yield strain of In_2Se_3 flakes is determined to be $\sim 5.5\%$ by atomic force microscopy (AFM) nano-indentation (see Supplementary Fig. 4 and Note 3), much larger than the domain-switching strain.

In addition, we have also applied tensile strain over four times the domain switching strain ($\sim 0.5\%$) on the ultrathin CVD-grown flakes by the two-point bending setup. As shown in Fig. R2 below, no fracture is observed as the uniaxial tensile strain reaches 2.15% , which is consistent with the derived $\sim 5.5\%$ breaking strain and confirms the achievable ferroelasticity in β' - In_2Se_3 below the breaking limit.

Figure R2. Optical images of the CVD-grown β' -In₂Se₃ upon relatively large uniaxial strain.

REVIEWERS' COMMENTS

Reviewer #1 (Remarks to the Author):

The authors have addressed all the questions, and I recommend its publication on Nature Communications.

Reviewer #2 (Remarks to the Author):

The authors have addressed my comments in a satisfactory manner. In particular, I am glad to see the new Raman experiments and thorough discussion on the electrostrictive effect. I can recommend its publication in Nature Communications now.

Reviewer #3 (Remarks to the Author):

In the revised version and the reply report, the authors have well answered all my comments. For my last round of questions, the author not only showed their detailed results and experimental details, but also pointed out the limits that can be achieved due to the limitations of experimental conditions. This work is sufficient novelty and the results are interesting and reliable. Therefore, I recommend it to be published in Nature Communications.

Response to Reviewers

Reviewer #1 (Remarks to the Author):

The authors have addressed all the questions, and I recommend its publication on Nature Communications.

RESPONSE: We thank the reviewer's recommendation for publication of the current manuscript in *Nature Communications*.

Reviewer #2 (Remarks to the Author):

The authors have addressed my comments in a satisfactory manner. In particular, I am glad to see the new Raman experiments and thorough discussion on the electrostrictive effect. I can recommend its publication in Nature Communications now.

RESPONSE: We thank the reviewer's positive comment on the new Raman and PFM results and recommendation to publish in *Nature Communications*.

Reviewer #3 (Remarks to the Author):

In the revised version and the reply report, the authors have well answered all my comments. For my last round of questions, the author not only showed their detailed results and experimental details, but also pointed out the limits that can be achieved due to the limitations of experimental conditions. This work is sufficient novelty and the results are interesting and reliable. Therefore, I recommend it to be published in Nature Communications.

RESPONSE: We appreciate the reviewer's positive comment and recommendation for publication of our work in *Nature Communications*.